# Emerging Regulatory Roles of Dual-Specificity Phosphatases in Inflammatory Airway Disease

**DOI:** 10.3390/ijms20030678

**Published:** 2019-02-05

**Authors:** Grace C. A. Manley, Lisa C. Parker, Yongliang Zhang

**Affiliations:** 1Department of Microbiology and Immunology, Yong Loo Lin School of Medicine, National University of Singapore, Singapore 117545, Singapore; micmgca@nus.edu.sg; 2Immunology Programme, Life Science Institute, National University of Singapore, Singapore 117597, Singapore; 3Department of Infection, Immunity and Cardiovascular Disease, University of Sheffield, Sheffield S10 2RX, UK; l.c.parker@sheffield.ac.uk

**Keywords:** inflammation, asthma, COPD, MAPK, respiratory viruses, influenza, rhinovirus, RSV

## Abstract

Inflammatory airway disease, such as asthma and chronic obstructive pulmonary disease (COPD), is a major health burden worldwide. These diseases cause large numbers of deaths each year due to airway obstruction, which is exacerbated by respiratory viral infection. The inflammatory response in the airway is mediated in part through the MAPK pathways: p38, JNK and ERK. These pathways also have roles in interferon production, viral replication, mucus production, and T cell responses, all of which are important processes in inflammatory airway disease. Dual-specificity phosphatases (DUSPs) are known to regulate the MAPKs, and roles for this family of proteins in the pathogenesis of airway disease are emerging. This review summarizes the function of DUSPs in regulation of cytokine expression, mucin production, and viral replication in the airway. The central role of DUSPs in T cell responses, including T cell activation, differentiation, and proliferation, will also be highlighted. In addition, the importance of this protein family in the lung, and the necessity of further investigation into their roles in airway disease, will be discussed.

## 1. Introduction

Inflammatory airway diseases are major causes of morbidity and mortality. The most common chronic respiratory diseases are asthma and chronic obstructive pulmonary disease (COPD), affecting around 300 million and 65 million people worldwide, respectively [1,2]. Both diseases are characterized by chronic inflammation of the respiratory tract, which is worsened in acute exacerbations, leading to airway obstruction, wheezing, and breathlessness [3]. The main cause of exacerbations is infection with respiratory viruses, including rhinovirus, respiratory syncytial virus (RSV), and influenza. Studies to determine the aetiology of exacerbations detected respiratory viruses in 65–82% of asthma exacerbations and 37–56% of COPD exacerbations [4,5,6,7,8,9,10,11].

The airway epithelium is the main target of respiratory viruses. Pattern recognition receptors (PRRs) on the surface and within epithelial cells recognize components of viruses and activate a range of signaling pathways, including the mitogen-activated protein kinase (MAPK) pathways [12,13]. The MAPK pathways consist of a three-tier kinase cascade, culminating in the dual-phosphorylation and activation of the MAPKs: extracellular signal-regulated kinase (ERK), Jun N-terminal kinase (JNK), and p38. These proteins translocate to the nucleus and activate a range of transcription factors, such as NF-κB and AP-1, leading to the production and release of many different molecules, including interferons, cytokines, and adhesion molecules [12,14], initiating inflammatory responses.

These responses are aberrant in patients with underlying airway disease. The reasons for this remain incompletely understood, but involve impaired control of viral infection [15,16], damaged epithelium [17,18], and altered lymphocyte responses [19,20]. This review will discuss the roles of the MAPK pathways in these processes and their regulation by a group of proteins known as dual-specificity phosphatases (DUSPs) or MAPK phosphatases (MKPs). 

## 2. The Epithelial Response to Respiratory Viral Infection

Activation of PRRs in respiratory epithelial cells leads to induction of the MAPK pathways, as summarized in Figure 1 [21]. Respiratory viral infection of epithelial cells can also activate the MAPKs through other means; for example, p38 can be activated by infection with rhinovirus, through the protein kinase Syk [22,23,24], or influenza, through the endoplasmic-reticulum stress response [25]. Once activated, the MAPKs have roles in many different processes, with severe implications in airway disease. These roles are summarized in the following sections. 

### 2.1. The MAPKs and Cytokine Release

The specific roles of each MAPK pathway have been examined using small molecule inhibitors. Pyridinyl imidazole compounds inhibit p38 by competing with ATP for its binding site, blocking its catalytic activity [26]. Griego et al. used two pyridinyl imidazole inhibitors, SB203580 and SB239053, to examine the role of p38 in cytokine and chemokine production by the BEAS-2B human bronchial epithelial cell line in response to infection with rhinovirus [27]. They found that infection caused a time- and dose-dependent increase in p38 phosphorylation. Treatment with either inhibitor prior to infection led to a significant reduction in the secretion of all cytokines and chemokines examined, including CXCL8, growth-related oncogene-α (GRO-α), granulocyte colony-stimulating factor (G-CSF), and granulocyte-macrophage colony-stimulating factor (GM-CSF), all of which have important roles in neutrophilia [27]. Recent work has furthered this knowledge, showing reduced production of CXCL8 by primary bronchial epithelial cells when p38 signaling was inhibited prior to infection with rhinovirus [28]. 

Inhibitors of p38 have also been used to highlight its importance in inflammatory cytokine production in response to other respiratory viruses. Treatment of A549 cells with SB203580 decreased release of CCL5 in response to RSV infection, and CXCL8 in response to parainfluenza virus infection [29,30]. Supporting this, inhibition of p38 in primary bronchial epithelial cells reduced mRNA production of IL-1β and TNF-α in response to RSV infection [31]. This pro-inflammatory role of p38 has also been demonstrated in vivo, as treatment of influenza-infected BALB/c mice with SB203580 lowered the concentration of TNF-α, IL-1β and IL-6 protein in lung homogenates [32]. 

The ERK pathway also has roles in cytokine induction in epithelial cells in response to viral infection. Liu et al. and Newcomb et al. treated airway epithelial cell lines with U0126 prior to rhinoviral infection. U0126 inhibits the ERK pathway by blocking activation of upstream kinases MEK1/2 [33]. Treatment with this drug reduced the secretion of CXCL8 in response to rhinovirus to almost baseline levels [34,35]; however, this was not replicated in primary bronchial epithelial cells treated with the MEK inhibitor PD90859 [28]. This could be due to differences in potency between the two chemical inhibitors, or between primary and immortalized cells. ERK signaling also induces inflammatory cytokine release in response to infection with RSV, with decreased levels of CXCL8 and CCL5 in supernatants of infected A549 cells treated with PD98059 [29,36]. 

Less is known about the role of the JNK pathway in inflammatory cytokine production in viral infection of the airway. One study showed weaker production and release of CXCL8 in response to infection with two strains of rhinovirus in primary bronchial epithelial cells treated with the JNK inhibitor SP600125 [28]. Together, these studies illustrate the central role of the MAPKs in the inflammatory response to respiratory viral infection. The precise contribution of each pathway seems to depend on the specific virus and cell type studied, but together they induce a large proportion of inflammatory cytokine production.

Respiratory epithelial cells release type I and type III interferons in response to viral infection (Figure 1) [37,38,39]. Interferons limit replication of respiratory viruses; pre-treatment of airway epithelial cells with interferon-β (IFN-β) significantly reduced replication of rhinovirus or influenza virus [40,41]. Several viruses, including influenza and RSV, target components of the interferon pathway in order to limit the antiviral response [42,43], and highly pathogenic strains of influenza induce lower levels of interferon [41]. The MAPK pathways have previously been implicated in interferon induction in response to influenza infection. Infection of MDCK cells or chicken macrophages with avian influenza viruses in the presence of JNK inhibitors led to increased viral replication due to decreased activation of IRF3 [44,45]. Recently, a gene expression array compared the response of primary HUVECs infected with highly pathogenic avian influenza viruses with and without SB202190, a p38 inhibitor. In addition to diminished production of inflammatory mediators, p38 inhibition reduced expression of IFN-β [46]. Signaling by ERK has also been linked to interferon signaling in RSV infection; ERK inhibition in A549 cells lessened activation of STAT1 in response to RSV [47]. This identifies the MAPKs as key pathways in both the anti-viral and pro-inflammatory responses to viral infection (Figure 1).

### 2.2. The MAPKs and Viral Replication

In addition to regulating respiratory viral infection through the interferon response, the MAPKs may also have roles in the viral life cycle. Marchant et al. showed that inhibition of p38 using SB203580 in the bronchial epithelial cell line 1HAEo- reduced replication of a number of respiratory viruses, including: influenza, RSV, coxsackie virus B3, human parainfluenza virus 3, and adenovirus [48]. Influenza genome replication occurs within the nucleus, forming viral ribonucleoprotein (vRNP) complexes, which are then exported into the cytoplasm [49]. Inhibition of either p38 or ERK was found to decrease influenza virus replication in MDCK cells due to a reduction in vRNP export from the nucleus [49,50,51]. Nencioni et al. hypothesized this was due to phosphorylation of vRNP by p38, affecting its affinity for the viral surface protein M1 [51]. This was supported by co-localization of p38 and vRNP in the nucleus of MDCK cells, and a reduction of vRNP phosphorylation when p38 was inhibited [51].

The roles of the MAPK pathways in RSV replication have also been investigated. Inhibition of p38 or ERK diminished levels of viral RNA and progeny release in A549 cells [52,53]. In both cases, this was thought to be due to impaired transport of viral proteins through the secretory pathway. Inhibition of p38 in vero cells decreased phosphorylation of the SH protein, a viral membrane protein with unknown function [54]. This altered the cellular distribution of SH, increasing localization in the golgi, implying that phosphorylation of SH may be necessary for transport through the secretory pathway and thus, viral assembly [54]. A similar role was proposed for ERK, as treatment of A549 cells with U0126 reduced surface expression of the viral F protein [53].

RSV and influenza viruses can also successfully evade the immune response and antiviral therapies by direct cell to cell spread [55,56]. RSV forms syncytia in the airway epithelium by fusing the membranes of neighbor cells, leading to cytosol mixing and viral transfer. RSV can also induce the formation of long filaments to reach, and spread to, more distant cells. This process is dependent on actin rearrangement through RhoA and the Arp2/Arp3 complex [56,57]. In wound healing assays, inhibition of ERK in epithelial cell lines reduced Arp2/3 recruitment and actin polymerization, indicating a possible role for ERK in syncytia formation during RSV infection [58]. ERK has previously been implicated in syncytia formation in cancer, with U0126 treatment of a choriocarcinoma cell line mitigating syncytia formation [59]. One result of syncytia formation in RSV infection is disruption of the epithelium and decreased membrane barrier integrity, which can lead to pneumonia and secondary bacterial infection. This can be modelled in A549 cells, where RSV infection lowers trans-epithelial resistance and causes paracellular gap formation. Treatment of A549 cells with SB203580 lessened these effects of RSV infection on the cell monolayer [60]. This was associated with reduced phosphorylation of heat shock protein 27 (Hsp27), a protein involved in actin rearrangement [60], suggesting that p38 may also be involved in syncytia formation through Hsp27.

As the MAPKs are involved in a wide variety of processes, they may also be indirectly involved in viral replication. For example, enteroviruses, such as rhinovirus, utilize the host cell endocytosis machinery, mainly the protein Rab11, to traffic cholesterol to replication organelles [61]. p38 has been shown to phosphorylate and activate guanyl-nucelotide dissociation inhibitor, a protein which facilitates cycling of Rab proteins between the membrane and the cytosol in endocytosis [62]. Cholesterol plays an important role in viral polyprotein processing and genome synthesis, and inhibition of cholesterol trafficking blocks viral replication [61]. Thus, p38 activation of Rab protein cycling may facilitate viral replication.

Overall, the literature suggests that respiratory viruses hijack the MAPKs and their downstream targets for their own ends; utilizing them for protein trafficking, viral assembly, and cell to cell spread. This highlights the need for strict regulation of these pathways, in order to limit viral replication, and proposes the MAPKs as targets for antiviral therapies [63].

### 2.3. The MAPKs and Mucus Production

A defining feature of asthma and COPD is goblet cell hyperplasia and excessive mucus production. This can lead to blockage of the airway and contributes to asthma-associated deaths [64]. The predominant mucin in asthma and COPD is MUC5AC [65,66]. The T helper 2 cytokine IL-13 is thought to be the main inducer of goblet cell hyperplasia and MUC5AC production in murine models of asthma, through activation of STAT6 [67,68]. The MAPKs also participate in this process; inhibition of p38 or ERK in differentiated primary murine or human airway epithelial cell monolayers reduced the IL-13 induced upregulation of goblet cell numbers and MUC5AC expression [69,70,71]. Furthermore, activation of p38 in response to IL-13 is weaker in epithelial cells from STAT6 knock out mice, indicating STAT6-induced mucin production occurs via p38 [69].

Respiratory viral infection has also been shown to upregulate mucus production. Double-stranded RNA is a common component or replication intermediate of viruses. Stimulation of NCI-H292 cells with double-stranded RNA upregulated expression of mucin MUC2, and this could be reversed by treatment with a p38 inhibitor [72]. MUC5AC expression is raised in ovalbumin murine models of asthma, and is increased further by RSV infection [73]. This observation may be dependent of IL-33, as IL-33 levels are higher in the lungs of RSV-infected mice, leading to enhanced production of IL-13. In addition, treatment of RAW cells with MAPK inhibitors decreases the release of IL-33 in response to RSV [74]. Inhibition of p38 has also been shown to repress IL-33 production in primary nasal epithelial cells in response to TNF-α stimulation [75].

Another mechanism by which mucin production is upregulated is via activation of the EGF receptor (EGFR) and Ras-Raf-MEK-ERK pathway [76,77]. Rhinovirus infection of differentiated primary human tracheal epithelial cells upregulates MUC5AC RNA levels and protein release, as well as RNA for MUC2, MUC3, MUC5B and MUC6 [78,79]. This induction was mediated by the EGFR pathway, as treatment with MEK or EGFR inhibitors returned MUC5AC levels to baseline. The authors hypothesized this was due to an autocrine loop, where rhinoviral infection induced production and release of EGRF ligands, as shown in NCI-H292 cells, which activated EGRF on the cell surface, and thus activated the ERK pathway [79]. This highlights the roles of the MAPKs in viral induced mucus production and has substantial implications for airway disease, where mucus hyperplasia is a common symptom.

### 2.4. Regulation of the MAPKs by DUSPs in Respiratory Viral Infection

The above studies underline the importance of the MAPK pathways in respiratory viral infection and airway disease. Although the majority of these studies rely on small molecule inhibitors which have significant off-target effects [80], they do indicate roles for the MAPKs in many of the processes implicated in exacerbations of asthma or COPD, including inflammation, mucus production and elevated viral replication. Thus, regulation of the MAPKs is of extreme importance. These pathways are primarily inactivated by simultaneous dephosphorylation of the threonine and tyrosine residues within the MAPK activation motif by dual-specificity phosphatases (DUSPs) (Figure 2) [81].

#### 2.4.1. DUSP1/MKP1

Much of the literature regarding DUSPs in innate immunity have focused on bacterial infection, and few studies have examined their roles in viral infection. DUSP1 (MKP1) is the archetype of the family and the most well studied. DUSP1 is a nuclear protein, capable of dephosphorylating p38, JNK and ERK, with substrate specificity depending on the stimuli and cell type [82,83]. It has been characterized as an early response gene, with undetectable expression at baseline, and rapid upregulation upon exposure to a variety of stimuli [82,84]. The airway epithelial cell line NCI-H292 upregulates DUSP1 mRNA within one hour in response to the synthetic double-stranded RNA molecule polyinosinic:polycytidylic acid (poly(I:C)) [85]. Poly(I:C) is a ligand for the PRRs toll-like receptor 3 (TLR3) and the RIG-I-like receptors (RLRs), which are predominantly activated by viral infection. Knock down of DUSP1 expression using small-interfering RNA (siRNA) in NCI-H292 cells amplified the release of two pro-inflammatory cytokines in response to poly(I:C) stimulation, CXCL8 and IL-6 [86]. A similar role for DUSP1 was seen in infection of the NCI-H1299 cell line with the avian coronavirus infectious bronchitis virus, with DUSP1 siRNA treatment increasing mRNA levels of CXCL8 in response to infection [87]. This augmented cytokine expression is likely to be due to elevated MAPK activation, with increased levels of phosphorylated p38 and JNK found in RSV infected A549 cells treated with DUSP1 siRNA [88].

DUSP1 has also been implicated in regulating the interferon response, with DUSP1 knock down in hepatocyte cell lines increasing STAT1 activation in response to hepatitis C virus or IFN-γ stimulation [89,90]. However, a yeast two-hybrid assay was unable to find an interaction between DUSP1 and STAT1, and overexpression of DUSP1 in COS-1 cells did not affect the level of STAT1 activation in response to IFN-γ [91]. Thus, the effects of DUSP1 knock down on the interferon response to hepatitis C infection may be indirect effects of increased MAPK activation (Section 2.1) rather than direct inactivation by DUSP1.

The inflammatory cytokine TNF-α is induced by respiratory viral infection, with higher expression in asthmatic patients [92,93], and elicits secondary inflammatory cytokine release from airway smooth muscle (ASM) cells [94]. Stimulation of primary ASM cells with TNF-α also caused the upregulation of DUSP1 mRNA and protein. When DUSP1 expression was knocked down in ASM cells, the release of CXCL8 increased in response to TNF-α stimulation [95]. CXCL8 is a neutrophil chemoattractant commonly detected in asthmatic airways [96]. Neutrophilia can harm the airway, causing epithelial cell damage and necrosis, and levels of CXCL8 correlate with asthma severity [96,97]. TNF-α stimulation of epithelial cells also induced the expression of mucins, and DUSP1 knock down in NCI-H292 cells further amplified the expression of airway mucin MUC5AC in response to TNF-α [98]. Taken together, this work suggests that DUSP1 has an important role in the response of the epithelium to insult, including regulation of inflammatory cytokine and mucin production.

#### 2.4.2. DUSP10/MKP5

DUSP10 (MKP5) is expressed ubiquitously in the nucleus and cytoplasm [99], and is upregulated in response to viral infection: bone-marrow derived macrophages (BMDMs) infected with influenza virus or stimulated with poly(I:C) have enhanced DUSP10 mRNA and protein expression [100]. Knock down of DUSP10 in primary bronchial epithelial cells increased the release of the neutrophil chemoattractants CXCL8 and CXCL1 in response to stimulation with a key proinflammatory cytokine IL-1β, suggesting that, like DUSP1, DUSP10 negatively regulates the inflammatory response in the airway [28]. Importantly, rhinoviral infection of airway epithelial cells or monocytes causes the release of IL-1β [28,101]; and combined stimulation with rhinovirus and IL-1β leads to an even greater inflammatory response in DUSP10 knock down primary bronchial epithelial cells from both healthy and COPD donors [28]. This identifies DUSP10 as a central regulator of the inflammatory response to respiratory viruses: infection of epithelial cells induces release of IL-1β, which acts back on the epithelium to promote inflammation, which is negatively regulated by DUSP10.

The role of DUSP10 in respiratory viral infection has also been examined in vivo: DUSP10 knock out mice infected with influenza had elevated levels of IL-6 in bronchoalveolar lavage (BAL) than wild-type mice. Interestingly, DUSP10 knock out mice also had decreased viral titres and better survival in response to infection. This was associated with raised expression and phosphorylation of IRF3, and therefore increased interferon (IFN) expression. Further investigation established that DUSP10 and IRF3 directly interact, indicating IRF3 as a novel substrate for DUSP10 and highlighting the importance of DUSP10 in regulating not only the inflammatory response, but also the anti-viral response.

Sustained, uncontrolled pulmonary inflammation can lead to acute lung injury, often seen in severe influenza infection. Murine models of acute lung injury can be generated by intratracheal injection of lipopolysaccharide (LPS), a TLR4 agonist. When DUSP10 knock out mice were utilized in an acute lung injury model, they exhibited greater disease severity than wild-type mice, with increased lung injury and pulmonary edema [102]. This was associated with augmented neutrophil influx in the lungs, and inflammatory cytokines in BAL. BMDMs isolated from these mice had elevated activation of p38 and JNK, and to a lesser extent ERK, in response to LPS treatment. Adoptive transfer of these BMDMs into wild-type mice led to enhanced lung inflammation in response to intratracheal LPS injection than the transfer of wild-type BMDMs [102]. This is in keeping with the in vitro data described above, and demonstrates that DUSP10 has an anti-inflammatory role in the airway, and is important in limiting immune-mediated lung damage.

#### 2.4.3. DUSP4/MKP2

Interestingly, one DUSP has been found to have a pro-inflammatory role in murine models of acute lung injury. In response to intratracheal LPS injection, DUSP4 (MKP2) knock out mice had decreased inflammatory cytokines in BAL and neutrophil infiltration in to the lung [103]. These data fit with an earlier study showing a pro-inflammatory role for DUSP4 in sepsis, with improved survival in DUSP4 knock out mice [104]. BMDMs taken from these mice produced lower levels of inflammatory cytokines in response to LPS injection than wild-type mice, associated with reduced activation of p38 and JNK, but increased activation of ERK. The authors suggest this was due to ERK-induced DUSP1 transcription, as has been demonstrated previously [105]. These studies indicate that different DUSPs may have pro- or anti-inflammatory roles in pulmonary inflammation. It should be noted that Al-Mutairi et al. found conflicting results, with DUSP4 knock out BMDMs releasing higher levels of inflammatory cytokines in response to LPS, although it is unclear why these studies differ [106].

## 3. T Cell Responses

Around 50% of asthma cases have an allergic phenotype, characterized by predominantly eosinophilic inflammation and T helper 2 (Th2) responses [19,107]. Higher levels of several Th2 cytokines have been found in BAL of asthmatics, including IL-4, IL-5, IL-13, IL-25, IL-33, and TSLP [108,109,110], and the levels of Th2 cytokines correlate with severity of asthma exacerbation [20]. The Th1/Th2 balance is also crucial for the immune control of respiratory viral infection. Asthmatics experimentally infected with rhinovirus had increased viral titres compared to infected healthy controls, with greater airway inflammation, bronchial hyperreactivity, and reductions in lung function associated with increased levels of IL-4, IL-5 and IL-13 in BAL [19]. The MAPKs have been implicated in induction of Th2 cytokines in the airway. Inhibition of p38 or ERK pathways in primary nasal epithelial cells or alveolar macrophages decreased release of IL-33 in response to TNF-α stimulation or RSV infection, respectively [74,75]. ERK and p38 inhibitors have also been used to confirm roles for these pathways in TSLP production in ASM cells in response to TNF-α or IL-1β stimulation [111]. Transcription of TSLP in ASM cells is also partially dependent on the AP-1 transcription factor, suggesting the involvement of JNK [112].

Early infection with RSV has been linked to the development of asthma, possibly through skewing the immune response towards a Th2 phenotype. Cytokine profiles of children infected with RSV revealed an expansion of Th2 cytokines and decreased Th1 cytokines [113], and RSV infection of mouse pups led to increased Th2 responses and impaired regulatory T (Treg) cell responses [114]. Enhanced T cell recruitment in RSV infection correlates with worsening symptoms [115], and ablation of either CD4^+^ cells or CD8^+^ cells in mouse models mitigates disease severity [116]. This Th2 skewing in response to infection may involve p38 MAPK. Infection of monocyte-derived dendritic cells with RSV induced expression of indoleamine-2,3-dioxygenase, an enzyme which favors Th2 differentiation by inducing apoptosis in Th1 cells. The expression of indoleamine-2,3-dioxygenase was reduced by treatment with the p38 inhibitor SB202190 [117].

Taken together, this work illustrates the importance of regulating the T cell response, and the T helper subset balance. In addition to affecting T cell activation through cytokine production by epithelial cells, the MAPKs have roles in T cells themselves, affecting their activation, proliferation and function [118].

### The Roles of DUSPs in T Cell Responses

Several studies have implicated DUSPs in the regulation of T cells and in the differentiation of T helper subsets (Table 1). DUSP1 knock out mice have been utilized to demonstrate roles for DUSP1 in T cell priming, proliferation, and T cell subset skewing. Antigen presenting cells (APCs), such as dendritic cells, have roles in this altered response. DUSP1 knock out dendritic cells had increased activation of p38, and thus altered cytokine production. This led to impaired priming of naïve wild-type T cells, with reduced differentiation into Th1 cells and augmented Th17 and Treg cell differentiation [119]. DUSP1 knock out T cells exhibit reduced proliferation in response to activation with anti-CD3 antibodies, and lower levels of IFN-γ and IL-17, Th1 and Th17 cytokines, respectively; while the Th2 cytokine IL-4 levels remained unchanged. These studies emphasize the different roles of DUSP1 in different cell types, with knock out having differing effects on APC mediated and T cell intrinsic responses. The overall effect of DUSP1 knock out was observed in influenza infection, with a decline in Th1 and CD8^+^ T cell numbers, leading to impaired control of the virus and greater disease severity. This altered response was associated with decreased nuclear translocation of NFATc1, a transcription factor important for T cell proliferation and function [120]. JNK was previously found to negatively regulate NFATc1 by phosphorylation [121], implying that the impaired T cell responses in DUSP1 knock out mice were due to increased JNK activation.

DUSP10 also plays a role in T cell proliferation. DUSP10 knock out mice have decreased numbers of virus specific CD4^+^ cells and CD8^+^ cells in the lung in response to influenza infection [100]. These mice also have diminished CD4^+^ cell proliferation in response to anti-CD3 and anti-CD28 antibodies; however, T cell effector functions were increased, with greater levels of Th1 and Th2 cytokine release. This elevated cytokine release was also observed in response to secondary infection with lymphocytic choriomeningitis virus, leading to immune-mediated death [122].

Three other DUSPs have also been implicated in T helper subset skewing: DUSP4, DUSP5 and DUSP16. DUSP4 negatively regulates Treg cell differentiation through inactivating STAT5 [123]. STAT5 is activated by IL-2, and is required for induction of Treg cells [124]. Overexpression of DUSP4 in HEK-293T cells reduced phosphorylation of STAT5 in response to IFN-β stimulation, and DUSP4 and STAT5 were co-immunoprecipitated, indicating that DUSP4 directly dephosphorylates STAT5. DUSP4 knock out mice had increased numbers of Treg cells and fewer Th17 cells [123]. In contrast to the negative regulatory role of DUSP4 in Treg cell generation, DUSP5 (hVHR3) seems to act as a positive regulator. Mice overexpressing DUSP5 had decreased inflammation and disease severity in a collagen-induced arthritis model, due to raised numbers of Treg cells and higher STAT5 activation, and reduced Th17 cells and lower STAT3 activation [125]. DUSP16 (MKP7) knock out shows embryonic lethality; however, mice expressing a dominant negative DUSP16 protein have been generated, and have an altered Th1/Th2 balance. T cells isolated from these mice produce increased levels of IFN-γ and diminished IL-4, IL-5 and IL-13 in response to anti-CD3 and anti-CD28 antibodies or ovalbumin [126,127]. In contrast to this, mice with a DUSP16 knock out specifically in the hematopoietic compartment do not display altered Th1 or Th2 responses, but demonstrate a role for the protein in regulating IL-2 production [128]. These mice had enhanced release of IL-2 and T cell proliferation, compared to wild-type mice, in response to anti-CD3 antibodies. This was associated with increased activation of ERK, which is critical for IL-2 expression [128]. IL-2 has previously been shown to inhibit the expansion of Th17 cells [129], and these mice exhibited a decrease in numbers of IL-17 producing cells, which was reversed by treatment with the ERK inhibitor U0126 [128].

In addition to their roles in T cell proliferation and T helper subset skewing, DUSPs have also been found to regulate T cell receptor (TCR) signaling. Binding of the TCR to antigen leads to recruitment of Lck, a tyrosine kinase, which phosphorylates the ζ chain, leading to ZAP-70 recruitment and the initiation of a range of signaling pathways [130]. Lck knock out mice emphasize its importance in the induction of TCR signaling [131]. T cells isolated from DUSP22 (JKAP) knock out mice had higher activation levels of several molecules downstream of the TCR, including Lck, ZAP-70, IKK, and the MAPKs, in response to anti-CD3 antibodies. DUSP22 and Lck were co-immunoprecipitated from murine splenic T cells, and DUSP22 was found to dephosphorylate Lck on Tyr394, inactivating it [132].

Further downstream of ZAP-70, TAB1 is activated by PKC-θ. TAB1 binds to TAK1, inducing an activating conformational change, triggering the IKK, p38 and JNK pathways. When DUSP14 (MKP-L) knock out T cells were stimulated with anti-CD3 antibodies, the numbers of activated CD69^+^ T cells were significantly higher than in wild-type T cells [133]. To investigate the reason behind this, HEK293T cells were transfected with wild-type or non-functional mutant DUSP14 and stimulated with anti-CD3 antibodies. Levels of activated Lck and ZAP-70 were unchanged between cells expressing wild-type and mutated DUSP14; however, IKK and MAPK activation was increased in cells expressing mutated DUSP14. Mass spectrometry was used to identify the binding protein and target of DUSP14 as TAB1, and further analysis revealed DUSP14 dephosphorylates TAB1 at the Ser358 residue [133].

These data illustrate that DUSPs have fundamental roles in adaptive immunity, affecting the activation, proliferation and differentiation of T helper cells. Although many of these studies have examined T cells in isolation, they identify the DUSPs as important regulators and suggest essential roles for them in airway diseases.

## 4. The Role of DUSP1 in Steroid Treatment

Exacerbations of asthma or COPD are treated with corticosteroids to limit the inflammatory response [134]. Steroids interact with the glucocorticoid receptor (GR) in the cytosol, inducing a conformational change, which allows the GR to translocate to the nucleus where it interacts with and inhibits transcription factors, such as NF-κB and AP-1 [135,136]. More recent evidence revealed that steroids mediate many of their actions through DUSP1. Treatment with the glucocorticoid dexamethasone increased DUSP1 expression in airway epithelial cell lines [137,138] and airway smooth muscle cells [139,140,141]. In mouse models, dexamethasone treatment reduced the release of inflammatory cytokines, TNF-α and IL-6, in serum in response to LPS injection. This inhibitory action of dexamethasone was weakened in DUSP1 knock out mice [142]. BMDMs or peritoneal macrophages isolated from DUSP1 knock out mice show that this was due to impaired inhibition of MAPK activation, and thus cytokine release, in response to LPS when DUSP1 is not present [142,143]. Dexamethasone treatment can also promote wound healing responses, upregulating proteins such as arginase 1 and fibroblast growth factors. This was diminished in peritoneal macrophages isolated from DUSP1 knock out mice in response to alternative macrophage activators IL-4 and IL-13, indicating that DUSP1 both restricts inflammation and promotes wound healing [144]. Interestingly, bone-marrow derived mast cells from DUSP1 knock out mice did not differ from wild-type in the levels of cytokines released in response to IgE cross-linking and dexamethasone treatment. This may be due to redundancy within the DUSP family, as dexamethasone also upregulated DUSPs 2, 4 and 9 [145]. This suggests the actions of DUSPs in dexamethasone treatment differ depending on cell type and stimulus. The role of DUSP1 in steroid treatment of the airway epithelium has also been investigated. siRNA knock down of DUSP1 in the A549 cell line blocked the anti-inflammatory action of dexamethasone on MAPK activation and cytokine release in response to IL-1β [146,147].

Around 10% of asthmatics are resistant to steroid treatment [148]. Several studies have examined the different responses between asthmatics who are sensitive to steroids, and those who are resistant. Bronchial biopsies from steroid-sensitive asthmatics show a reduction in JNK activation after treatment with dexamethasone, which was not seen in bronchial biopsies from steroid-resistant asthmatics [149]. Higher activation levels of p38 were also detected in alveolar macrophages isolated from steroid-resistant asthmatics than in steroid-sensitive cells. This was associated with reduced expression of DUSP1 in response to dexamethasone [150]. Lower expression of DUSP1 was also found in peripheral blood neutrophils from steroid-resistant asthmatics in comparison to steroid-sensitive asthmatics [151].

Exacerbations of asthma and COPD caused by viral or bacterial infection are also more resistant to steroid treatment than non-viral exacerbations [152,153]. Rhinovirus infections impair the anti-inflammatory actions of steroids, partly by reducing nuclear translocation of the glucocorticoid receptor [154]. Treatment of A549 cells with dexamethasone reduced inflammatory cytokine release and upregulated DUSP1 in response to IL-1β stimulation. However, when these cells were infected with rhinovirus, this suppression by dexamethasone was abrogated, as was the upregulation of DUSP1 expression [154]. Treatment of A549 cells with the TLR2 ligand Pam_3_CSK_4_ also induced steroid resistance, but had no effect on DUSP1 expression. However, Pam_3_CSK_4_ treatment did induce oxidative stress, and a proportion of the DUSP1 present in these cells was oxidized [155]. Oxidation of the catalytic cysteine residue in the active site of DUSPs renders them inactive [156]. These findings suggest that steroid insensitivity in asthmatics, or in infected airways, may be due to a defect in DUSP1 expression or activation. Furthermore, polymorphisms in the DUSP1 gene have been associated with steroid responsiveness [157]. The roles of other DUSPs in steroid treatment remain to be investigated.

Another therapy commonly used to treat asthmatics are bronchodilators, such as β_2_-recpetor agonists. In addition to bronchodilation, β_2_-agonists also have some anti-inflammatory effects. Treatment of cells with β_2_-agonists increases intracellular levels of cAMP [158]. The promoter of DUSP1 contains a cAMP-response element [159], and β_2_-agonist treatment of airway epithelial cell lines and airway smooth muscle cells has been found to upregulate DUSP1 expression [160,161]. The role of DUSP1 in the anti-inflammatory action of the β_2_-agonist salbutamol was investigated by inducing paw edema in wild-type and DUSP1 knock out mice. Salbutamol treatment reduced the level of inflammation by around 70% in wild-type mice, but only by around 40% in DUSP1 knock out mice [137]. This signifies that DUSP1 also has an important role in mediating the anti-inflammatory effects of β_2_-agonists, in addition to corticosteroids.

## 5. Studies Linking DUSPs to Asthma and Sarcoidosis

It has been suggested that several of the DUSPs may be dysregulated in people with asthma. This is an intriguing explanation for the excessive inflammatory response seen in these patients, particularly because activation of the MAPK proteins is elevated in asthmatics. Baseline levels of phosphorylated p38 are higher in bronchial epithelial cells isolated from asthmatics than healthy controls, and phosphorylated ERK is greater in T cells isolated from asthmatics both at baseline and in response to anti-CD3 antibodies [162,163]. A study in 2008 isolated nasal epithelial cells from healthy individuals and patients with house dust mite allergy, a common allergy associated with asthma. They performed a microarray to determine any gene expression changes in response to stimulation with house dust mite. In non-allergic controls, DUSP1 mRNA expression was upregulated in response to house dust mite challenge; however, in allergic patients, DUSP1 expression did not alter [164]. Altered expression and activation of p38 and DUSP1 has also been observed in sarcoidosis. Sarcoidosis is a systemic inflammatory disorder, often characterized by granulomas within the lung. Rastogi et al. isolated and cultured leukocytes from the BAL of controls and sarcoidosis patients [165]. Activation levels of p38 and inflammatory cytokine production were higher in cells isolated from patients, both at baseline and after PRR stimulation. This coincided with much lower DUSP1 protein expression in controls than sarcoidosis patients [165]. These findings suggest impaired DUSP1 upregulation as a reason for enhanced inflammatory responses in asthma and sarcoidosis.

Cigarette smoke exposure can lead to the development of asthma and is the most common cause of COPD [166]. This is partially through the induction of inflammatory responses in the lung, with higher levels of inflammatory cytokines detected in the lungs of smokers [167]. Higher levels of p38 activation have also been detected in smokers’ lungs [168], and treatment of BEAS-2B cells with p38 inhibitors reduced the release of cytokines in response to cigarette smoke [169]. DUSP1 may also have a role in this process, as ferrets exposed to cigarette smoke for six months had reduced levels of DUSP1 protein in lung tissue, although the functional effects of this have yet to be examined [170].

DUSP10 may also be differentially expressed in asthmatic patients. A transcriptional profile of Th2 cells taken from asthmatic and healthy subjects showed lower baseline mRNA expression of DUSP10 in the asthmatic Th2 cells than the healthy cells [171]. Intriguingly, a single nucleotide polymorphism in the DUSP4 gene was identified in a genetic screen for variants associated with severe asthma. However, this was not statistically significant, possibly due to the limited number of patients in the study [172]. Studies examining the relationship of other DUSPs and asthma would be of interest and are yet to be carried out.

## 6. Conclusions

The MAPK pathways have important roles in airway inflammation and are aberrantly activated in several inflammatory airway diseases. This may in part be due to altered expression of DUSPs, with lower baseline levels of DUSP10 and lower induction of DUSP1 expression upon allergen stimulation or steroid treatment in asthmatics. DUSPs have central roles in regulating inflammation, therefore, this aberrant expression could have important functions in the pathogenesis of lung inflammation. Underlying airway disease also leads to greater susceptibility to lower respiratory tract infections, due to impaired control of viral replication. The literature discussed here suggest a possible role for DUSPs in controlling viral-induced exacerbations of airway disease, not only in terms of regulating the MAPKs and their roles in viral life cycles, but also IFN production, T cell proliferation and Th2 skewing. Current treatments of airway inflammation are not always effective and cause significant side effects. Therefore, the development of new, more specific, treatments is of extreme importance. MAPK inhibitors have been investigated for therapeutic application with varying success [173]. An alternative method of reducing inflammation would be via upregulation of DUSPs. Mechanisms by which this may be achieved have been reviewed previously [174]. DUSPs represent potential targets for novel anti-inflammatory treatments of airway disease and future work into their roles in the airway is imperative.

## Figures and Tables

**Figure 1 ijms-20-00678-f001:**
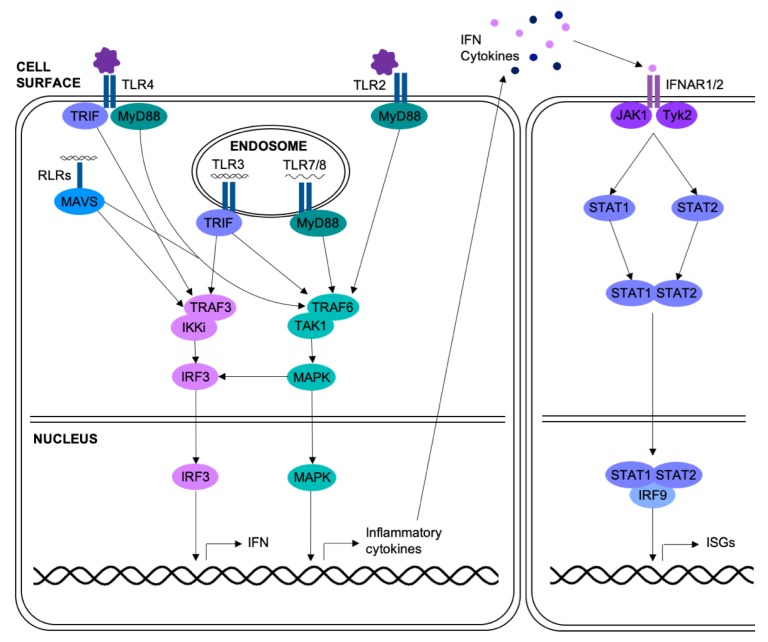
Activation of signaling pathways in respiratory epithelial cells upon viral infection. PRRs detect viral infection of the cell: TLRs 2 and 4 can bind components of the viral surface, TLR3 binds dsRNA, TLR7/8 bind ssRNA, and the RLRs bind dsRNA or 5′-triphosphorylated ssRNA. Adaptor proteins MyD88, TRIF, and MAVS mediate the activation of signaling pathways, including the MAPK pathways. The MAPKs translocate into the nucleus where they activate transcription factors, leading to the transcription of genes for inflammatory cytokines. TRIF and MAVS signaling activates IRF3, leading to interferon production. The MAPK pathways can also activate IRF3. Inflammatory cytokines and interferons are released by the cell and act upon surrounding cells. IFN binds to the IFN receptor complex IFNAR1/2, activating the JAK/STAT pathway. JAK1 and Tyk2 phosphorylate STAT1 and STAT2 which dimerize, translocate to the nucleus and bind IRF9, forming ISGF3, which induces transcription of interferon stimulated genes (ISGs).

**Figure 2 ijms-20-00678-f002:**
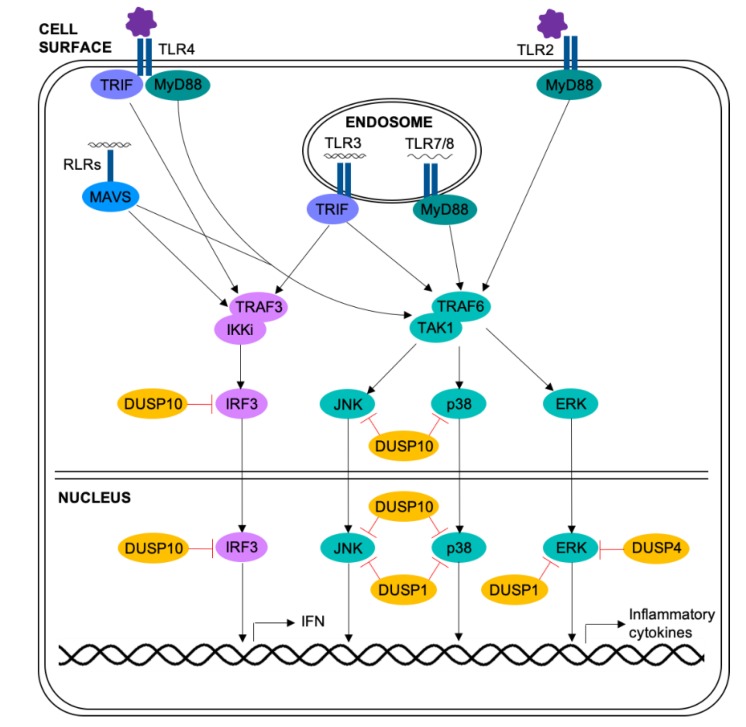
Regulation of the MAPK pathways by DUSPs in epithelial cells upon viral infection. PRR recognition of viruses or viral components activates the MAPK and IRF3 pathways. The MAPKs and IRF3 translocate to the nucleus and induce expression of inflammatory cytokines and interferon. These pathways are negatively regulated (red arrows) through dephosphorylation by DUSPs. DUSP1 is present in the nucleus and dephosphorylates all three MAPKs. DUSP4 is a nuclear protein, and is thought to dephosphorylate ERK. DUSP10 is present in both the nucleus and the cytoplasm and dephosphorylates JNK, p38 and IRF3. Black arrows represent activating interactions, red arrows represent inhibition.

**Table 1 ijms-20-00678-t001:** Roles of DUSPs in T cells.

DUSP	Regulates Proliferation	Regulates TCR Signaling	Regulates Subset Differentiation				Reference
			Th1	Th2	Th17	Treg	
DUSP1	X		Promotes		Promotes	Inhibits	[120]
DUSP4					Promotes	Inhibits	[123]
DUSP5					Inhibits	Promotes	[125]
DUSP10	X						[122]
DUSP14	X	X					[133]
DUSP16	X		Inhibits	Promotes	Promotes		[127,128]
DUSP22	X	X					[132]

* Blank boxes = not determined.

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
