# Peer review of "Emerging Regulatory Roles of Dual-Specificity Phosphatases in Inflammatory Airway Disease"

_ijms, 2019, doi:10.3390/ijms20030678_

Reviewer 1 Report

The review article presented a comprehensive analysis on the role of DUSPs/MKPs in airway diseases.  It is informative and will be of great interest to the readership of the researchers in the field. The article is well written and important literatures are cited.  I only have some relative minor comments with some statement, terminologies, and expressions.

Major issue:

Page 3, line 67: Figure 1: The implication of dephosphorylation of STAT1 by DUSP1 is problematic.  The evidence presented in the cited paper is weak at best, and can be explained by alternative mechanisms.  Although an earlier paper (Venema et al.J. Biol. Chem.273:3079530800, 1998) provided evidence to suggest dephosphorylation of STAT1 by DUSP1/MKP-1, a later investigation by Steve Keyse's group provided compelling evidence against direct dephosphorylation of STAT1 by DUSP1 (Slack et al. J Biol Chem. 276(19):16491-500, 2001).

Given what was described in ref 119 that more Th17 and less Th1 cells were found in mice with knockout Mkp-1 (Mkp-1 KO APC), it might be helpful to clarify the message expressed in Table 1.  The authors might want to think how to reconcile the seemingly contradictory findings, from the angle of T-cell-intrinsic or APC-mediated?).

The conclusion section spent three sentences on asthmatics, without mentioning any other diseases.  Other diseases were also discussed in earlier sections. There is a lack of symmetry in the conclusion section. Perhaps other diseases discussed in the earlier sections should also be briefly touched to present a bird view on the broad scope of airway diseases.

Minor issues:

Page 1 line 40: A reference should be cited to support the statement of the regulation of NF-kappaB by MAPKs.  It is less obvious how MAPK directly regulates NF-kappaB.

Page 2, line 49: Change "figure 1" to "Figure 1".

Page 3, Line 92-3: U0126 is far more potent than PD90859.  This may also contribute to the difference.

Page 5, line 116: Please use either STAT1 or STAT-1, but not both.

Page 5, line 128: Add "by" after "supported".

Page 5, line 189: Should it be MEK inhibitors?

Page 6, line 220: It might be better to add a word “secondary” after “elicits” to clarify that these are cytokines induced by TNF.

The authors might want to think to divide section 2.4. to different paragraphs with a sub-title such as 2.4.1 MKP-1/DUSP1, 2.4.2 DUSP10/MKP-5, DUSP4 (MKP-2), etc.  

Page 8 line 324, page 10 line 385: “DUSPs 4, 5, and 16” are unconventional use.  The authors might want to use DUSP4, DUSP5, and DUSP16. 

Page 9, line 378: Another reference (Wang et al.  Life Sciences 83: 671–680, 2008) should also be cited, because more information in vivo and in vitro were presented.

Page 10, second paragraph: Defects in MKP-1 and p38 were also seen in patients with sarcoidosis.  Perhaps this should also be mentioned in this paragraph.

Page 10, line 410: Perhaps the statement “This is supported by the fact that …….” is too strong, since it is unclear how the SNPs affect DUSP1 expression or activity. 

Page 11, line 434:  It might be better to change “This proposes…” to “These findings suggest…..”.  Line 436: Cigarette smoking was raised here, but did not mention how MAPK or MKPs are involved.  This topic should be expanded or omitted.

Page 11, line 441: “asthmatic cells” is a too non-specific term. It would be helpful to specify what kind cells they refer to. Did the article deal with T cells, epithelial cells, or mast cells? 

Page 11, line 441-443: The sentence “ Intriguingly…..” is difficult to read. Please restructure it.

Author Response

We thank the reviewer for their kind feedback and helpful comments.

Major issues:

1.      Page 3, line 67: Figure 1: The implication of dephosphorylation of STAT1 by DUSP1 is problematic.  The evidence presented in the cited paper is weak at best, and can be explained by alternative mechanisms.  Although an earlier paper (Venema et al.J. Biol. Chem.273:30795–30800, 1998) provided evidence to suggest dephosphorylation of STAT1 by DUSP1/MKP-1, a later investigation by Steve Keyse's group provided compelling evidence against direct dephosphorylation of STAT1 by DUSP1 (Slack et al. J Biol Chem. 276(19):16491-500, 2001).

Response: Thank you for these references. The regulation of STAT1 by DUSP1 has been removed from Figure 1 (page 2). The text has also been amended. The paragraph discussing DUSP1 regulation of STAT1 has been moved from section 2.4.2 to 2.4.1 (see point 11), and is now at lines 226-232. Two sentences have been added to make it clear that these are controversial findings: “However, a yeast two-hybrid assay was unable to find an interaction between DUSP1 and STAT1, and overexpression of DUSP1 in COS-1 cells did not affect the level of STAT1 activation in response to IFN-gamma [91]. Thus, the effects of DUSP1 knock down on the interferon response to hepatitis C infection may be indirect effects of increased MAPK activation (Section 2.1) rather than direct inactivation by DUSP1.”

2.      Given what was described in ref 119 that more Th17 and less Th1 cells were found in mice with knockout Mkp-1 (Mkp-1 KO APC), it might be helpful to clarify the message expressed in Table 1.  The authors might want to think how to reconcile the seemingly contradictory findings, from the angle of T-cell-intrinsic or APC-mediated?).

Response: The first paragraph in section 3.1 (page 8) has been altered to make these seemingly contradictory findings clearer. The description of reference 119 has been expanded and moved earlier on, so that the description of T cell priming comes before T cell proliferation and differentiation (lines 320-324). An additional sentence has been added to emphasize the different findings in each cell type (lines 326-328).

3.      The conclusion section spent three sentences on asthmatics, without mentioning any other diseases.  Other diseases were also discussed in earlier sections. There is a lack of symmetry in the conclusion section. Perhaps other diseases discussed in the earlier sections should also be briefly touched to present a bird view on the broad scope of airway diseases.

Response: Thanks for the suggestion. The conclusion has been altered to be less specific to asthma and to discuss lung disease in general and the potential for therapeutic intervention.

Minor issues:

4.      Page 1 line 40: A reference should be cited to support the statement of the regulation of NF-kappaB by MAPKs.  It is less obvious how MAPK directly regulates NF-kappaB.

Response: We apologize, this was simplified for clarity. We were referring to ERK and p38 activating MSK1/2 which in turn phosphorylates the p65 subunit of NF-kappaB, increasing transcriptional activation. Reference 14 (Vermeulen et al. The EMBO Journal, 22(6), pp.1313-1324, 2003) has been added to make this clearer (now line 41).

5.      Page 2, line 49: Change "figure 1" to "Figure 1".

Response: This has been altered.

6.      Page 3, Line 92-3: U0126 is far more potent than PD90859.  This may also contribute to the difference.

Response: A sentence has now been added to reflect this. “This could be due to differences in potency between the two chemical inhibitors, or between primary and immortalized cells.” Line 91-92.

7.      Page 5, line 116: Please use either STAT1 or STAT-1, but not both.

Response: Thanks for pointing out this. We have now removed the hyphen (now line 114) and checked through the rest of the document to ensure it is consistent.

8.      Page 5, line 128: Add "by" after "supported".

Response: This has been added in, now line 126.

9.      Page 5, line 189: Should it be MEK inhibitors?

Response: Yes, this has now been altered to MEK inhibitors, now line 187.

10.  Page 6, line 220: It might be better to add a word “secondary” after “elicits” to clarify that these are cytokines induced by TNF.

Response: This has now been added, line 234.

11.  The authors might want to think to divide section 2.4. to different paragraphs with a sub-title such as 2.4.1 MKP-1/DUSP1, 2.4.2 DUSP10/MKP-5, DUSP4 (MKP-2), etc. 

Response: Thanks for the suggestion. These subsections have now been added in, lines 209, 244 and 276.

12.  Page 8 line 324, page 10 line 385: “DUSPs 4, 5, and 16” are unconventional use.  The authors might want to use DUSP4, DUSP5, and DUSP16.

Response: This has now been altered, lines 340-341.

13.  Page 9, line 378: Another reference (Wang et al.  Life Sciences 83: 671–680, 2008) should also be cited, because more information in vivo and in vitro were presented.

Response: This important reference has now been included (reference 142) in the following sentences “In mouse models, dexamethasone treatment reduced the release of inflammatory cytokines, TNF-alpha and IL-6, in serum in response to LPS injection. This inhibitory action of dexamethasone was weakened in DUSP1 knock out mice [142]. BMDMs or peritoneal macrophages isolated from DUSP1 knock out mice show that this was due to impaired inhibition of MAPK activation, and thus cytokine release, in response to LPS when DUSP1 is not present.” Lines 392-396.

14.  Page 10, second paragraph: Defects in MKP-1 and p38 were also seen in patients with sarcoidosis.  Perhaps this should also be mentioned in this paragraph.

Response: Thank you for drawing our attention to this article. This has now been included in section 5, lines 451-458, and sarcoidosis has been added to the section subtitle (line 441).

15.  Page 10, line 410: Perhaps the statement “This is supported by the fact that …….” is too strong, since it is unclear how the SNPs affect DUSP1 expression or activity.

Response: This has now been altered to “Furthermore,...” (line 428).

16.  Page 11, line 434:  It might be better to change “This proposes…” to “These findings suggest…..”. 

Response: This has been altered, now line 457.

17.  Line 436: Cigarette smoking was raised here, but did not mention how MAPK or MKPs are involved.  This topic should be expanded or omitted.

Response: This has now been expanded and forms a separate paragraph (lines 459-465).

18.  Page 11, line 441: “asthmatic cells” is a too non-specific term. It would be helpful to specify what kind cells they refer to. Did the article deal with T cells, epithelial cells, or mast cells?

Response: We apologize, earlier in the paragraph the cells are referred to as Th2, but it was unclear whether these were the same cells in this sentence. This has now been rectified by including “Th2” in line 468.

19.  Page 11, line 441-443: The sentence “ Intriguingly…..” is difficult to read. Please restructure it.

Response: We apologize, this sentence was confusing. It has now been changed to “Intriguingly, a single nucleotide polymorphism in the DUSP4 gene was identified in a genetic screen for variants associated with severe asthma.” Line 468-469.

Reviewer 2 Report

This is a well written, detailed and comprehensively referenced review on the role of DUSPs in inflammatory airway disease. It covers multiple relevant human conditions including asthma, allergy and infection, particular those caused by viruses. The sections related to the role of the MAPK pathways and regulation by DUSPs in cell-mediated immunity is particularly important and well dealt with.

This is a timely review in light of the intriguing role these MAPK regulators play in modulating disease. It is likely to be of wide appeal to both scientific and clinical immunopathologists.

I have a few recommendations:

The title should avoid abbreviations if possible

Figure 1 is excellent but appears too early in the manuscript. For example, DUSPs are introduced more fully in section 2.4 but Figure 1 (which refers to DUSPs) appears at the start of section 2. I can see 2 ways of dealing with this – perhaps have an earlier short introduction to DUSPs more broadly, or move Figure 1 (and refer to it more frequently)

The following DUSPs are mentioned: DUSP1, 4, 5, 10, 14, 16 and 22 but it is not always clear what the functional difference is between the DUSPs. Can this possible be included in a table or figure?

Section 2.1 deals with some pharmacological attempts to block the MAPK pathways. Are there any new interventions or clinical trials looking at targeting MAPK pathways via DUSPs that could be mentioned towards the end of the manuscript?

Author Response

We thank the reviewer for their kind feedback and helpful comments.

1.      The title should avoid abbreviations if possible

Response: Thanks for the suggestion. The title has now been altered from DUSPs to Dual-Specificity Phosphatases.

2.      Figure 1 is excellent but appears too early in the manuscript. For example, DUSPs are introduced more fully in section 2.4 but Figure 1 (which refers to DUSPs) appears at the start of section 2. I can see 2 ways of dealing with this – perhaps have an earlier short introduction to DUSPs more broadly, or move Figure 1 (and refer to it more frequently)

Response: Figure 1 has now been replaced with a diagram showing the activation of signaling pathways by viral infection to ensure the signaling pathways discussed in the text are clear, but without the inclusion of DUSPs (page 2, line 54). Additional references throughout the text have been made to Figure 1 (lines 102-103 and 116). An additional figure describing the roles of DUSPs has been included later on in the document, page 6, where it is more relevant in the revised manuscript.

3.      The following DUSPs are mentioned: DUSP1, 4, 5, 10, 14, 16 and 22 but it is not always clear what the functional difference is between the DUSPs. Can this possible be included in a table or figure?

Response: Thanks for pointing out this. Figure 2 has been added to help explain the functional differences between DUSPs 1, 4 and 10 in epithelial cells, including their subcellular localization and molecular targets (page 6, line 201). DUSPs 5, 14, 16 and 22 are only discussed in terms of T cell responses, and the functional differences are presented in table 1 (page 10, line 383) in the revised manuscript.

4.      Section 2.1 deals with some pharmacological attempts to block the MAPK pathways. Are there any new interventions or clinical trials looking at targeting MAPK pathways via DUSPs that could be mentioned towards the end of the manuscript?

Response: The conclusion has been expanded to include discussion of the potential for therapeutic interventions targeting DUSPs, referencing previous reviews which have covered this topic (lines 483-487).